# Limiting Factors that Influence the Formation of Producer Groups in the South-East Region of Romania: A Fuzzy Set Qualitative Comparative Analysis (fsQCA)

**Andrei-Mirel Florea** [1,*][iD]**, Alexandru Capatina** [2]**, Riana Iren Radu** [2]**, Constanța Serban (Bacanu)** [1]**, Madalina Georgiana Boboc** [1]**, Cristina Stoica (Dinca)** [1]**, Mihaela Munteanu (Pila)** [1]**, Iuliana Manuela Ion (Dumitriu)** [1] **and Silvius Stanciu** [3,*][iD]

[1]  The School for Doctoral Studies in Engineering, "Dunarea de Jos" University of Galati, 800001 Galati, Romania; serbancing@yahoo.com (C.S.B.); mada91mada@yahoo.com (M.G.B.); dincacristina2007@yahoo.com (C.S.D.); mihaela_pila@yahoo.com (M.M.P.); Iuliana.Ion12@yahoo.com (I.M.I.D.)

[2]  Departament of Business Administration, Dunarea de Jos University/Faculty of Economics and Business Administration, 800001 Galati, Romania; alexcapatina@gmail.com (A.C.); raduriana@gmail.com (R.I.R.)

[3]  Center for Technology Transfer, "Dunarea de Jos" University of Galati, 800001 Galati, Romania

\*  Correspondence: floreaandreim@yahoo.com (A.-M.F.); sstanciu@ugal.ro (S.S.)

**Abstract:** The fragmentation of the Agricultural Real Estate in Romania, which is due to the lack of vision regarding the retrocession of the land and to the ineffective measures for reparcelling farmland, manifests by the existence of a record number of about 3422 million farms in Romania, of which about 92% have a utilized agricultural area (UAA) below 5 ha. The Romanian agricultural sector possesses about 30% of the total European farms but contributes only 3% to the total EU agricultural production. The association of local agricultural producers may be an alternative to reparcelling farmland on a short-term basis in order to reduce the fragmentation degree and increase the competitiveness of the national agricultural sector. According to the statistics of the Ministry of Agriculture and Rural Development, 25 active groups of agricultural producers are registered in the South-East Region in 2018, where these associative entities have been recorded to have a low degree of viability. The paper proposes an analysis on the farmers' motivation regarding the access to/exit from a form of agricultural association and the identification of some alternatives for increasing the viability of the associative forms in the agricultural sector in the South-East of Romania. In this respect, a study was carried out on a sample of 16 entities that gave up their status of producer groups in the analyzed region in 2011–2018. The Fuzzy Set Quantitative Comparative Analysis (fsQCA) Qualitative analysis method was applied, which identified the main influence factors that have led to the disappearance of this associative form. We consider this study relevant for drawing attention to the main obstacles that Romanian farmers face in joining an associative form. The study has shown that mainly the factors directly influenced by government policies have led to the withdrawal of a relatively large number of producers from producer groups in the studied region.

**Keywords:** Romania; agriculture; association; fsQCA; producer groups

## 1. Introduction

The practice of subsistence farming through small agricultural organizations, characterized by low productivity, limited access to resources, and inefficient use of machinery, is one of the main

causes of reduced competitiveness in the Romanian agricultural sector [1]. A major contribution to the achievement of the current farm structure in Romania was brought by the measures of restoring the property rights initiated in the early 1990s [2]. The importance shown at the level of the agriculture sector and the observance of the Common Agricultural Policy established by the European Union have led to the intensive reformation process of both agriculture and rural communities [3]. The restructuring of agriculture is a slow process, but there is an improvement in what concerns land consolidation manifested by the diminishment of small farms and by the growth of those over 10 hectares [2].

According to data published by the National Institute of Statistics (NIS) [4], the number of agricultural associations decreased by 20% in the year 2016, mainly due to the 8.9% reduction of the farms administering areas smaller than 1 ha. The reparcelling of the agricultural land was based on association, lease, or acquisition of some lands, inclusively by shareholders with foreign capital [5].

The Ministry of Agriculture and Rural Development, [6] through the National Program for Rural Development (NPRD), is helping farmers to overcome the manifested malfunctions and advocates for establishing some associative forms in order to gain a solid market position. Stakeholder collaboration is essential to collectively achieve a competitive advantage over the complex sustainability requirements in the agricultural food, environmental, business, and social sectors [7]. By working together, the competitiveness of the agricultural sector will increase, with potential positive implications for industrial food processing at the national level [8]. Following the promotion of such measures, it is important to understand the factors that determine or prevent individuals from taking joint actions and the type of organizational structures they choose [9].

Specialty literature presents us with a variety of factors that limit the association of producers [6,9,10], but we aim at proving that the non-adjustment of the governmental policies regarding the association of producer groups to the current situation affects the other involved factors, a fact which is characteristic of Romanian agriculture as a whole, irrespective of the region. From this perspective, we consider that the producers have already understood the necessity of association, but governmental policies on associative forms in agriculture are not optimized in the sense of drawing the producers into a form of association.

Although the national agricultural programs and the programs funded by European funds support and encourage the initiative of association by providing facilities for agricultural producer associations, the government fiscal measures cancel out the interest in these facilities. Through the State's fiscal policy, the producer groups are taxed twice [11]. Moreover, the access to non-reimbursable funding from European Funds is conditional on the volume of marketed production [12], which implies a forced taxing of agriculture. It is well known that a fairly large proportion of Romanian agriculture is not taxed [13]. The low volume of marketed production combined with poor agricultural practices leads to a low level of non-reimbursable funds drawn in agriculture, which leads to the orientation of small producers towards the development of individual agricultural exploitation.

The Fuzzy Set Quantitative Comparative Analysis (fsQCA) method can be applied for the study performance, as it allows the combined analysis of the causes that determined a certain outcome [14]. This method is recommended when the research includes conditions that are sufficient (but not necessary) to determine the outcome [15]. The usage of the fsQCA method will generate a complex solution that will present the way in which factors interact in order to produce the result.

## 2. Theoretical Background

### 2.1. Factors that Allow the Collaboration

The need for association and cooperation has been manifested in the agricultural sector, more than in any other field [16]. In particular, small firms lack the financial resources, the competence, and the legitimacy that could enable them to reach potential customers [17]. The low levels of productivity and competitiveness in agricultural production units can be partially explained by the lack of knowledge

regarding new technologies and by insufficient interaction among local actors, leading to a low level of technology adoption and the failure to adopt new or improved practices in the area [18]. According to Fałkowski et al. [9], closer cooperation between agricultural producers is often considered to be an appropriate way of building the competitiveness of agricultural enterprises. According to Ruben and Heras [19], the good functioning of a collective farm is dependent on mutual trust and on the engagement relationship between members in order to achieve the common goals, see Figure 1.

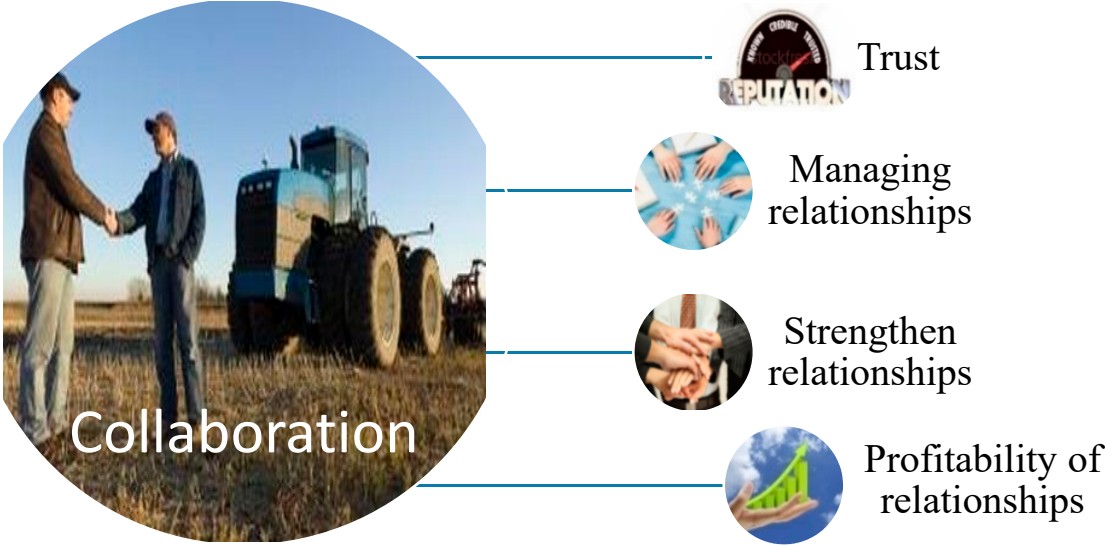

**Figure 1.** The basis for the creation and continuity of the collaboration relationship. Source: Own contribution.

Moreover, the relations between members, once created, must be carefully managed with the aim of maintaining and strengthening loyalty by aligning the interests of the members as individual entities with those of the cooperative [20].

The performance of the actions developed within the group, the efficient management of the relationships and their profitability, through the fulfillment of the established measures, determines the continuation of the collaboration [21,22].

On the other side, if the objectives that were set up when establishing the associative form have not been achieved or one of the members or the whole organization is subjected to internal and external pressures, it may lead to the loss of group identity [23].

One of the factors that stimulates the performance of collaboration in the formation and functioning of the supply chain is represented by the maintenance of a stable environment, which will generate long-term benefits for the whole system [7]. The role of government policies is crucial in supporting the creation, maintenance, and development of associative forms, see Figure 2. The possibility of obtaining financial resources and tax benefits, as a result of sustainable support and development policies, is another key factor determining the cooperation between farmers [24,25].

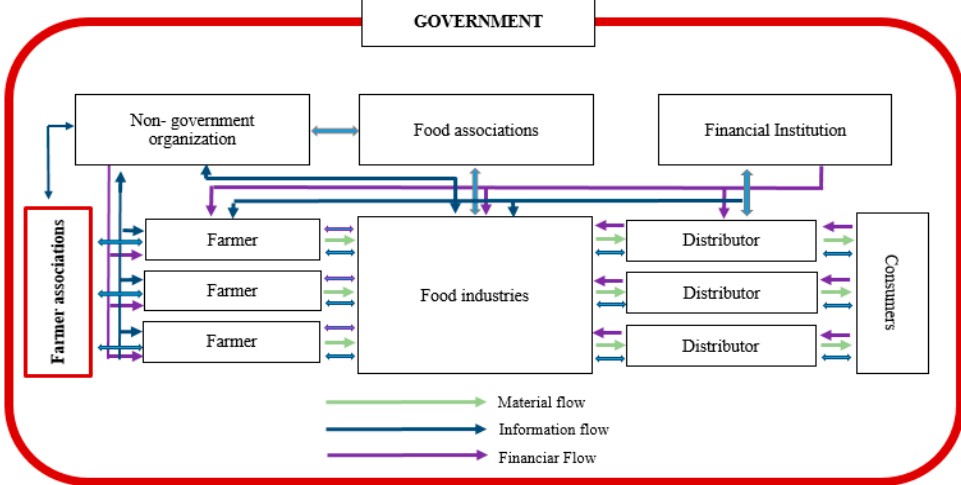

**Figure 2.** The general representation of an agri-food supply chain. Source: Adapted from Dania et al. [7].

## 2.2. Status of Romanian Producer Groups

The forms of co-operation of producers differ in terms of the organizational structure applied and the value and scope of these activities [26]. The cooperation system encountered in Romanian agriculture is at an early stage of diversification by types of collective farms [27]. The main types of associative forms encountered are: Producer groups and collective farms [28]. Cooperation through producer groups is regulated in Romanian legislation by GO 37/2005. According to this legal act, the main role of producer groups is represented by the joint marketing of individual farmers' products.

In order to be recognized as a producer group according to Government Ordinance no. 37/2005, the following conditions must be fulfilled:

➢ to consist of at least five members who are farmers and/or forest holders;
➢ to have a centralized system of accounting, billing, registration and quantitative, qualitative, and value tracking of the members' production;
➢ to hold qualified personnel, appropriate to the product group and/or products for which it is recognized or to contract specialized services;
➢ to have in the constitutive act and/or statute provisions on the obligations of the members.

For the recognition of the agricultural producer capacity, according to the provisions of Law no. 145/2014 and Law 170/2017, only a producer certificate issued by the local authorities and a marketing certificate are required.

The transition from a simplified marketing system to an association type of system requires major changes in the organization of the management and accounting of the marketed production, accompanied by passing to other grids/categories for profit/loss taxation, which is more difficult to be borne by the producer.

Luca and Toderiţa [11] state that the possible benefits of the association appear to be rather medium- and long-term, which implies that producer members have to cope with the burdens induced by changing the work style (recognition of the group being the first of them) in the early phase of the association. Starting from this idea, we can say that the producers who sell only on the basis of a producer's certificate issued by local authorities and a marketing certificate are clearly favored compared to those who decide to register as a producer group.

According to the measures implemented through the Common Agricultural Policy (CAP), the associative forms have a high priority in accessing European funds. However, this advantage is turned into a disadvantage in Romanian agriculture due to the doubtful quality and the lack of vision of the projects in conjunction with an excessive bureaucracy [29].

For example, accessing European funds for setting up producer groups (measure 142 for the 2007–2013 period became measure 9.1 for 2014–2020 period) did not present a major interest for producers. Luca and Toderiţa [13] considered that the main reason would be the fact that some promoted measures had, as the inspiration source, the model of some countries with a developed agriculture, without taking into account the fact that much of Romanian agriculture is not taxed.

Analyzing the provisions of the Guide on sub-measure 9.1 [12], we note that accessing this measure influences the level of attracted funds, as follows:

(a) "the subsidy (5% in the first two years) is related to the sold production", so it must be justified by accounting documents;
(b) "group evaluation is related to the sale of 75% of production through the group", again proven by supporting documents.

The study conducted by Hernández-Espallardo et al. [30] confirms the hypothesis that the members of an association remain united as long as this collaboration offers them more development opportunities than individual ones.

In this sub-measure, for agricultural producers or, in the present case, for the members who renounced the status of producer group, the incentives granted through the financing measures were not sufficient to lead to its establishment or maintenance.

## 3. Materials and Methods

The information needed to carry out the study was collected from the National Institute of Statistics (NIS), the Ministry of Agriculture and Rural Development (MARD) database, and from the survey conducted by 16 producer groups from the SE Region of Romania.

The representatives of the groups were contacted by phone to determine how the factors identified in the conceptual model influenced their decision to withdraw from the producer group.

The bibliographic documentation was made by consulting the Clarivate Analytics, Scopus, and Google Academic scientific databases.

In order to analyze the factors obtained from the interview, the Fuzzy Set Quantitative Comparative Analysis (fsQCA) method was applied. The use of this method is particularly useful in order to highlight the way in which the analyzed factors interact with each other [31] and in order to perform pertinent analyses on smaller samples.

FsQCA [32] is a set-theoretical technique in which X and Y represent the calibrated member scores set. The fsQCA formulates a qualitative requirement that the condition X is required for the Y result, see Figure 3. This formula assumes the application of the entire range of X and Y values. The FsQCA does not express membership as a necessary condition. This type of technique is ideal for this study for two fundamental reasons: On the one hand, to analyze not only the isolated effect of two or more variables on the result of interest but also to explore all the possible interactions between these variables. The other aspect concerns the sample size. The advantage of this method is that fsQCA allows researchers to work with medium-sized samples without the need to get a large amount of individual cases [31].

Using consistency as a factor that determines the correlation of factors and coverage as a measure of covering them in producing the outcome, the fsQCA provides accurate information on how causal configuration may generate a particular outcome [32].

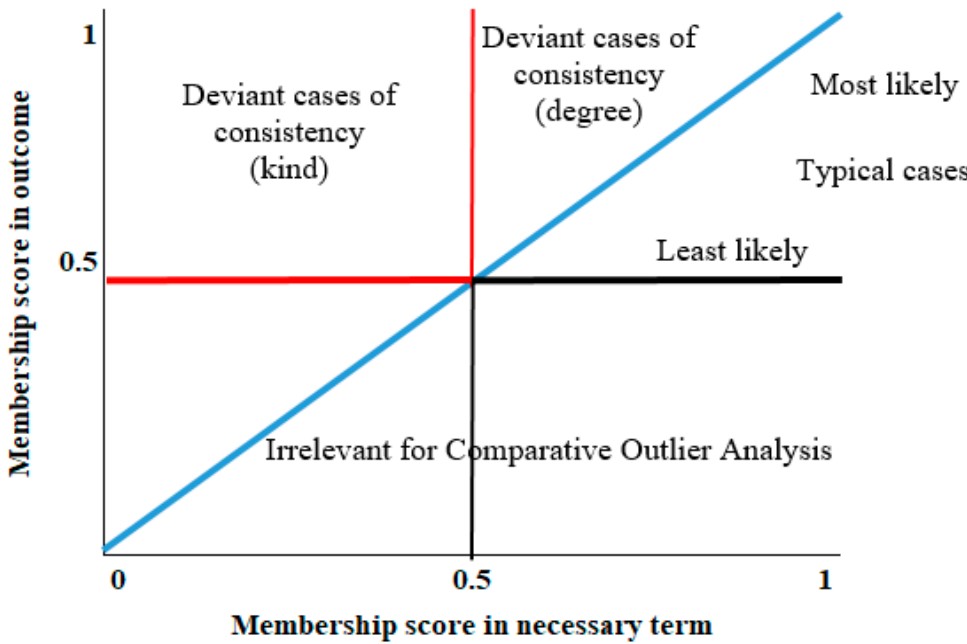

**Figure 3.** Type of cases in Fuzzy Set Quantitative Comparative Analysis (fsQCA) (necessity). Source: Adapted from Nair and Gibbert [33].

Following from studying specialty literature in conjunction with the results of the analysis carried out at the sample level, five factors (poor access to funds, unprepared person, legislative framework, misunderstandings between members, and bankruptcy) which led to the abandonment of the producer group status in the studied region were identified.

The identification of factors started from the hypothesis according to which European funds can be accessed more efficiently and easily through the producer groups. In the thematic publication edited under the aegis of the Ministry of Agriculture and Regional Development [10], a series of factors which affect the association at the level of Romanian agriculture have been identified. Among these factors, only those that can determine the defection from the group structure have been selected and adapted by the authors. These factors have been tested later on within the defined sample.

The first identified factor was represented by the fact that the fiscal policy, noted in the conceptual model as the "legislative framework", was not readjusted [13].

These legislative changes, a result of government policy, had a direct and indirect negative impact on the absorption rate of European funds, with the individual producers or producer groups having to make significant efforts to benefit from funding [11].

The "unprepared person" factor included the issue of a shortage of qualified personnel for executing high-performance agricultural activities and the lack of vision in the writing of applications [29].

The joint management, corroborated with the changes which affected the drawing in of funds, has led to the occurrence of tensions which have generated the third factor of the respective model, misunderstanding members [21]. The possibility of management which did not succeed in optimally managing the problems and consolidating the relations within the group [22] has also been taken into consideration within the conceptual model. The adverse management was only included within the conceptual model because its inclusion within the questionnaire was not considered opportune due to the fact that the interviewed persons tend to manifest a subjective opinion with regard to acknowledging or not acknowledging their inefficient management. The hypothesis according to which the shortfall of the owned funds, as well as the influence of the other factors, has led to a debt degree which cannot be managed, was taken into consideration. The term bankruptcy was defined

with this purpose, and this term was included within the model. The conceptual model of the factors involved, as well as the representative bodies legislating these factors, are presented in Figure 4.

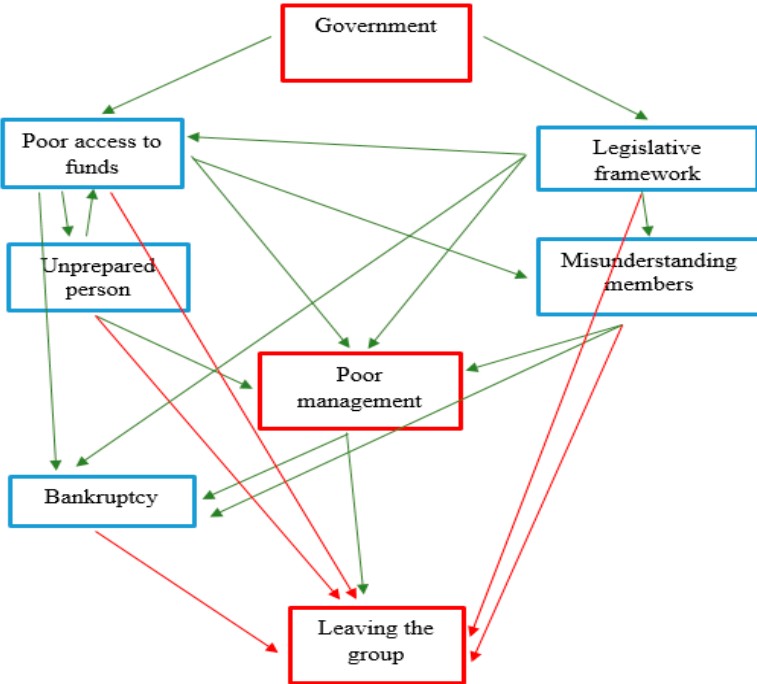

**Figure 4.** The model of factors affecting the integrity of producer groups. Source: Own contribution.

### 3.1. Data Collection

The study was carried out covering the South-East Region of Romania, a region that spans an area of 35,762 km$^2$, representing 15% of the total area of the country, see Figure 5, and is the second largest region of Romania from a physical point of view [34]. Within this region, all landforms are present, the plain surface being the predominant landform.

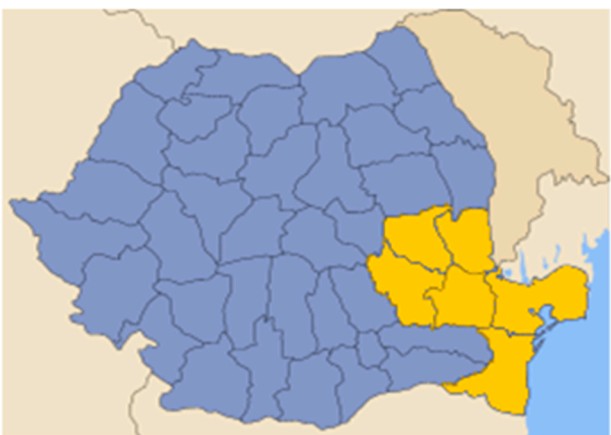

**Figure 5.** Location of the case study—Romania's South-East Development Region. Source: Adapted from the South East Regional Development Agency website.

The survey was carried out on a sample of 16 former members of producer groups that had withdrawn from this form of organization during the period 2011–2018 [35].

Respondents were encouraged to include an assessment of the influence of each factor in their response using the Likert five-point scale presented in Table 1. At the same time, respondents were encouraged to say whether they were influenced by factors other than those identified by the author.

### 3.2. Calibration Process

According to Ragin [36], the QCA method identifies a combination of conditions that generate a result, without excluding the possibility of multiple combinations of conditions that can produce the same result.

In addition to identifying the unique conditions required, the fsQCA also identifies the necessary combinations of conditions and distinguishes between the necessary AND necessary OR combinations. Based on this premise, a fuzzy set calibration approach was used in order to determine the influence and relation of inhibitory factors in maintaining producer groups. Each factor has been replaced with values between 0 and 1 [37], depending on the answers given by the interviewees. The scale used to calibrate inhibitory factors is shown in Table 1.

**Table 1.** Calibration of scales.

| Scale Point | Fuzzy-Set Value |
|---|---|
| Strongly influenced | 1 |
| Significant influence | 0.75 |
| Influenced | 0.5 |
| Insignificant influence | 0.25 |
| Unaffected | 0 |

Source: Own contribution.

By means of the fsQCA software, a new variable called "giving up cooperation" was created, which was presented in Table 2, obtained by compiling the conditions from the conceptual model:

Giving up cooperation = fuzzyand (bankruptcy, misunderstandings between members, poor access to funds, legislative framework, unprepared persons)

Taking into account our research context, an analysis of variance (ANOVA) would add value and deepen the analysis. Given the data sample with their associated estimation procedures (such as the "variation" among and between groups), ANOVA would help to analyze the differences across the sample.

Further development of the paper should observe the hypothesized relationships depending on the inhibitory factors that determine the withdrawal from the associative forms through the development of a model conceived through Structural Equation Modelling (SEM). This approach will involve a structural model that imputes relationships between latent variables related to the configurations of causal factors presented in this study. Thus, it would diminish the risk of generalizing the paper's results.

**Table 2.** Calibration of all variables.

| Case | Bankruptcy | Misunderstood Members | Poor Access to Funds | Legislative Framework | Unprepared Persons | Leaving Group | Giving Up Cooperation |
|---|---|---|---|---|---|---|---|
| 1 | 0.25 | 1 | 0.75 | 0.75 | 0.25 | 1 | 0.25 |
| 2 | 0.25 | 1 | 1 | 1 | 1 | 1 | 0.25 |
| 3 | 0 | 0.5 | 0.75 | 0.75 | 0.25 | 1 | 0 |
| 4 | 0 | 0 | 0.75 | 0.25 | 0 | 1 | 0 |
| 5 | 0 | 0.5 | 0.5 | 0.75 | 0.5 | 1 | 0 |
| 6 | 1 | 0.75 | 0.25 | 0.25 | 0.25 | 1 | 0.25 |
| 7 | 0 | 0.25 | 0.75 | 0.75 | 0.75 | 1 | 0 |
| 8 | 0 | 0.25 | 1 | 0.25 | 0.25 | 1 | 0 |
| 9 | 0 | 0 | 1 | 1 | 0 | 1 | 0 |
| 10 | 0.25 | 0.75 | 0.75 | 0.75 | 0.5 | 1 | 0.25 |
| 11 | 1 | 0 | 0 | 0.25 | 0 | 1 | 0 |
| 12 | 0 | 1 | 1 | 1 | 0.5 | 1 | 0 |
| 13 | 0 | 0 | 1 | 1 | 0 | 1 | 0 |
| 14 | 0 | 0 | 0.5 | 0.75 | 0 | 1 | 0 |
| 15 | 0 | 0 | 0.75 | 0.25 | 0.25 | 1 | 0 |
| 16 | 0 | 0 | 1 | 0.5 | 0 | 1 | 0 |

Source: Processed by the author with the fsQCA 3.0 software.

## 4. Findings

In the inception part of the analysis, it was investigated whether the identified causes were necessary and sufficient to obtain the result, i.e., leaving the producer group. In this analysis, it should be noted that the cause is necessary to produce the result, but the presence of the cause does not always ensure the presence of the result [38].

According to Schneider et al. [39], a 0.9 consistency threshold should be recorded in the causality assessment.

The next step was to perform a sufficiency analysis from which the consistency measure was derived, which should be greater than 0.75 [31]. Sufficiency means that the cause (X) can produce the result (Y), but the result can be produced by other causes, too [38].

The measurement of necessity and sufficiency was done with the help of the truth table, see Table 3, and of the graphical representation highlighted in Figure 6. According to the graphical representation of the cases in the XY Plot graph, it results that all the conditions under analysis are necessary and sufficient by positioning them above the diagonal of the graph.

The consistency score is 1, whereas the coverage score is 0.0625. These scores imply that the distribution of fuzzy sets is largely consistent with the assertion that giving up cooperation is a subset of the result of leaving the group.

**Table 3.** Truth table analysis for the research sample. Source: Processed by the author with the fsQCA 3.0 software.

| Bankruptcy | Misunderstanding Members | Poor Access to Funds | Legislative Framework | Unprepared Person | Number | Leaving Group | Raw Consist. | PRI Consist. | SYM Consist |
|---|---|---|---|---|---|---|---|---|---|
| 0 | 0 | 1 | 0 | 0 | 3 | 1 | 1 | 1 | 1 |
| 0 | 0 | 1 | 1 | 0 | 2 | 1 | 1 | 1 | 1 |
| 1 | 0 | 0 | 0 | 0 | 1 | 1 | 1 | 1 | 1 |
| 1 | 1 | 0 | 0 | 0 | 1 | 1 | 1 | 1 | 1 |
| 0 | 1 | 1 | 1 | 0 | 1 | 1 | 1 | 1 | 1 |
| 0 | 0 | 1 | 1 | 1 | 1 | 1 | 1 | 1 | 1 |
| 0 | 1 | 1 | 1 | 1 | 1 | 1 | 1 | 1 | 1 |

The researchers understand the causal formulae by using a "Truth table" [32]. The rows of the truth table present all the logical combinations of causal conditions (equal to 2x, where x equals the number of causal conditions). The truth table is the number of cases that are involved in that special causal combination. In other words, according to Ragin [36], "the truth table elaborates and

formalizes one of the key analyses, strategies of comparative research—the examination of the of specific communication cases, combinations of causal conditions, to see if they have the same results".

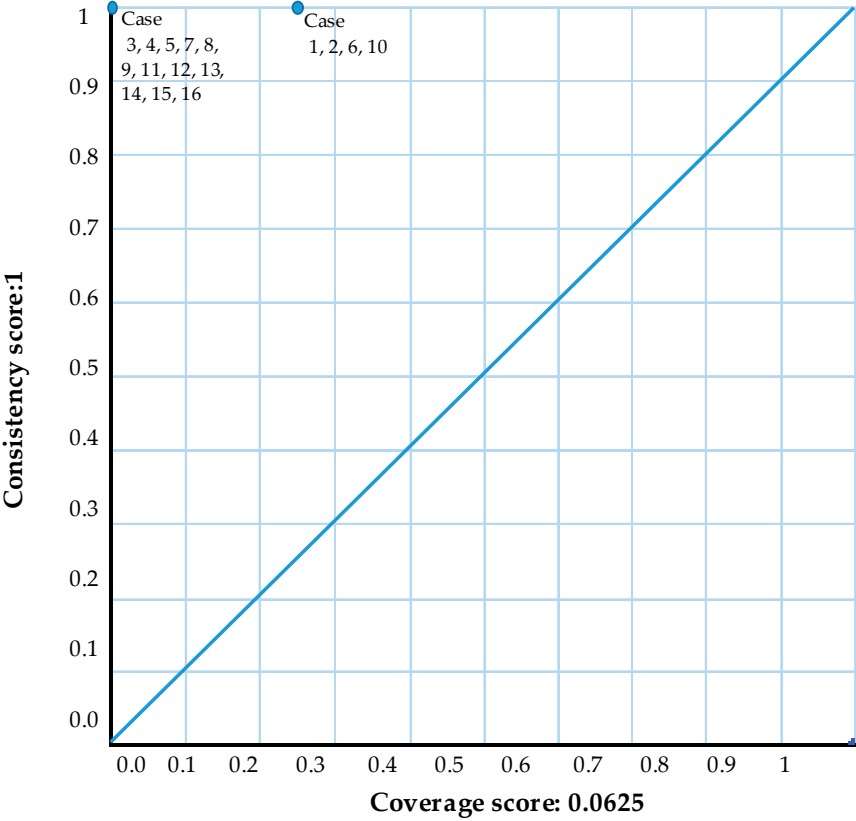

**Figure 6.** Distribution of cases within XY Plot graph. Source: Processed by the author with the fsQCA 3.0 software.

The truth table represents the main tool for understanding causal relationships. From the moment the truth table is drawn, causal combinations are analyzed through two factors: Consistency and coverage [37]. Seven configurations were identified and are presented in Table 3.

The coherence is the degree to which the data supports the established theoretical claim advocated by the researcher (i.e., the necessity or sufficiency). For sufficiency, consistency, it indicated the degree in which a cause (or a causal formula) is actually a subgroup of the result [31]. After running the Quine-McCluskey algorithm (~bankruptcy* poor access to funds* legislative framework), the complex solution that is shown in Table 4 was generated:

**Table 4.** Complex solution.

| Complex Solution | Raw Coverage | Unique Coverage | Consistency |
|---|---|---|---|
| ~bankruptcy*poor access to funds*legislative framework | 0.5625 | 0.21875 | 1 |
| bankruptcy*~pooraccesstofunds*~legislativeframework*~unpreparedperson | 0.125 | 0.09375 | 1 |
| ~bankruptcy*~misunderstandingbetweenmembers*pooraccesstofunds*~unpreparedperson | 0.453125 | 0.125 | 1 |
| Solution coverage: 0.78125 | | | |
| Solution consistency: 1 | | | |

Source: Processed by the author with the fsQCA 3.0 software.

The combination of a low level of bankruptcy and a high level of poor access to non-reimbursable funds, together with the instability of the legislative framework, represents a necessary condition for an individual to give up group membership. The influence of this condition on the result is presented in Table 5.

**Table 5.** Analysis of necessary conditions. Outcome variable: Leaving group.

| Conditions Tested | Consistency | Coverage |
| --- | --- | --- |
| ~bankruptcy + poor access to funds + legislative framework | 0.875000 | 1.000000 |

Source: Processed by the author with the fsQCA 3.0 software.

The tested condition reveals that in an environment where high access to funding cannot be ensured and there are legislative changes related to access criteria, producer groups defect from joint activities. The bankruptcy factor at a negative level reveals that this element has no significant influence. A high level of bankruptcy would lead not only to leaving the status of a producer group but to the abandonment of agricultural activities. As Ertimur and Venkatesh [21] stated, the key to viable cooperation is the degree of performance of the actions and their profitability.

The political decisions elaborated for the purpose of regulating Romanian agriculture were also studied by Ciutacu et al. [40], who state that these decisions lead to an irreversible decline and increase in the gap with other EU members. Ciutacu et al. [40] consider that there is a need for a fundamental change in both agricultural policies and programs, but also in the rural development that is applied synergistically and continuously.

Another problem, namely that of accessing European funds, is also dealt with by Bogza et al. [41], highlighting that the objectives included in business plans by specialized consultancy firms are sometimes unrealistic, uncorrelated with frequent legislative changes, or difficult to implement. A similar opinion is expressed by Lucian [29], who states that a lack of vision in project development is one of the main causes of a low fund absorption rate in the agricultural sector.

The importance of implementing European reforms and adapting them to the needs of Romanian agriculture by supporting the market for the sale and development of competitive farms is also presented by Popescu et al. [1], stating that these actions represent important support measures for small- and medium-sized farms.

Since the domain-specific literature mainly deals with the factors that prevent the formation of an associative form, the present study aims at completing the literature with the inhibitory factors that determine the withdrawal from the associative forms. These factors are the antithesis with those involving association. Our opinion is similar to that expressed by Hernández-Espallardo et al. [30] according to which members of an association remain united as long as cooperation gives them more advantages than individual actions.

## 5. Conclusions

Practicing a policy that supports associative forms, both at the European Union level and at a national level, has stimulated the desire for cooperation among the producers.

The need for such a change has been understood by some producers who have joined such associations with the aim of reducing costs, risks, and increasing the visibility of the agri-food market.

The study has shown that mainly the factors directly influenced by government policies have led to the withdrawal of a relatively large number of producers from producer groups in the studied region.

In Romania, the changes in the financing framework of producer groups and the limitation of financing in relation to the marketed production represented obstacles to development. The fluctuating market influenced by factors such as political instability, high bureaucracy, lack of adequate outlets, and limited resources, has led to a macroeconomic context which has strongly affected the associative forms of agricultural producers.

The results of the study support the hypothesis of non-compliant policies developed at the level of agriculture with influences on the producers in the studied area. In only the last two years, five producer groups have withdrawn after a collaboration between 5 and 10 years.

Nevertheless, the study is limited because it is performed only in one of the regions of the country, a region that still has a significant share in Romanian agriculture. External factors that prevent South

East Region producers from performing joint actions, given the legislative framework, are similar to those across the country.

However, to ensure high accuracy of the study, we intend to extend the study of these factors to another region to reconfirm the conclusions of the present one in a subsequent article.

**Author Contributions:** A.-M.F. writing—original draft preparation, A.C., A.-M.F. and C.S.B. conceived and designed methodology, S.S. and R.I.R. writing—review and editing, A.-M.F., M.G.B, C.S.D. investigation and analyzed the data, A.-M.F., M.M.P., I.M.I.D. visualization.

**Funding:** This research was funded by The School for Doctoral Studies in Engineering.

**Acknowledgments:** The authors would like to thank the editors and the anonymous reviewers of this journal. What is more, the authors express great thanks for the financial support from The School for Doctoral Studies in Engineering.

**Conflicts of Interest:** The authors declare no conflict of interest.

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
