# Peer review of "Limiting Factors that Influence the Formation of Producer Groups in the South-East Region of Romania: A Fuzzy Set Qualitative Comparative Analysis (fsQCA)"

_sustainability, doi:10.3390/su11061614_

Round 1
Reviewer 1 Report
An interesting article referring to an important and current topic related to horizontal integration in agriculture. Written in an interesting and accessible way, supported by well-chosen literature of the subject. However, the slight drawback is its technical / editorial underlining, which I mention below.
There could be added at least one or two sentences about the obtained results to the abstract.
For Keywords, I suggest adding “Producer Groups”.
Figure 2 - indistinct, I suggest correcting.
2.1. Factors that allow the collaboration (Good practices on collaboration / association) - I suggest removing this fragment in brackets -> (Good practices on collaboration / association)
No paragraph indents in lines: 105, 130, 196
Verses 120, 201, 208, 248 are unnecessarily empty
Figure 3. Please improve the quality
Figure 4. Conceptual model (please clarify the title, the title should be more precise)
Line 249 sentence "The scale used to calibrate factors in Table 1." unnecessarily starts with a new line, I suggest moving it to line 247
Table 3. Please improve the readability of the headers
Figure 5, 6 lack of source
Table 1, 2, 3 lack of source
Author Response
What needs to be reviewed | What I've reviewed | |
1 | There could be added at least one or two sentences about the obtained results to the abstract. | The necessary changes have been made to the article |
2 | For Keywords, I suggest adding “Producer Groups”. | The necessary changes have been made to the article |
3 | Figure 2 - indistinct, I suggest correcting. | The necessary changes have been made to the article |
4 | 2.1. Factors that allow the collaboration (Good practices on collaboration / association) - I suggest removing this fragment in brackets -> (Good practices on collaboration / association) | The necessary changes have been made to the article |
5 | No paragraph indents in lines: 105, 130, 196 | The necessary changes have been made to the article |
6 | Verses 120, 201, 208, 248 are unnecessarily empty | The necessary changes have been made to the article |
7 | Figure 3. Please improve the quality | The necessary changes have been made to the article |
8 | Figure 4. Conceptual model (please clarify the title, the title should be more precise) | The necessary changes have been made to the article |
9 | Line 249 sentence "The scale used to calibrate factors in Table 1." unnecessarily starts with a new line, I suggest moving it to line 247 | The necessary changes have been made to the article |
10 | Table 3. Please improve the readability of the headers | The necessary changes have been made to the article |
11 | Figure 5, 6 lack of source | The necessary changes have been made to the article |
12 | Table 1, 2, 3 lack of source | The necessary changes have been made to the article |
Reviewer 2 Report
The introduction, the background and the methodology provided sufficient background. The description is quite clear. Sound findings supported the study conclusions. Some minor suggestions are made here for further amending for the authors.
The format of the references in the list needs to be check again.
In the lines 383 and 384 in Table 3, there are two blue blocks to demonstrate "editing the true table". The authors are suggested to add a note to clearly present their purpose to use the blue block.
Please explain why the authors added line 234-235 in the article. It seems promptly appear in the context.
In line 44-49, the published years of the literature cited is not needed, according to the format applied in this journal.
Author Response
What needs to be reviewed | What I've reviewed | |
1. | The format of the references in the list needs to be check again. | The bibliography has been revised in accordance with the requirements in the Author Guidelines |
2. | In the lines 383 and 384 in Table 3, there are two blue blocks to demonstrate "editing the true table". The authors are suggested to add a note to clearly present their purpose to use the blue block. | In the fsQCA 3.0 program in the truth table field, that line remained selected. We have replaced the truth table without leaving any selected square. |
3. | Please explain why the authors added line 234-235 in the article. It seems promptly appear in the context. | Adding that line was made for a better understanding. This has been moved to the materials and methods section |
4. | In line 44-49, the published years of the literature cited is not needed, according to the format applied in this journal. | The necessary changes have been made to the article |